# The Prediction Accuracy of Digital Orthodontic Setups for the Orthodontic Phase before Orthognathic Surgery

**DOI:** 10.3390/jcm11206141

**Published:** 2022-10-18

**Authors:** Olivier de Waard, Frank Baan, Robin Bruggink, Ewald M. Bronkhorst, Anne Marie Kuijpers-Jagtman, Edwin M. Ongkosuwito

**Affiliations:** 1Department of Dentistry-Orthodontics and Craniofacial Biology, Radboud University Medical Center, 6500 HB Nijmegen, The Netherlands; 2Radboudumc 3DLab, Radboud Institute for Health Sciences, Radboud University Medical Center, 6500 HB Nijmegen, The Netherlands; 3Department of Dentistry, Radboud Institute for Health Sciences, Radboud University Medical Center, 6500 HB Nijmegen, The Netherlands; 4Department of Orthodontics, University Medical Center Groningen, University of Groningen, 9713 GZ Groningen, The Netherlands; 5Department of Orthodontics and Dentofacial Orthopedics, School of Dental Medicine, Medical Faculty, University of Bern, CH-3010 Bern, Switzerland; 6Faculty of Dentistry, Universitas Indonesia, Jakarta 10430, Indonesia

**Keywords:** orthodontics, CBCT, orthognathic surgery, setup

## Abstract

The purpose of this study was to verify whether pre-treatment digital setups can accurately predict the tooth positions after presurgical orthodontic treatment has been performed in a 3-dimensional way. Twenty-six patients who underwent a combined orthodontic-orthognathic surgical treatment were included. Pre-treatment digital dental models were merged with cone beam computed tomography (CBCT) scans. One operator fabricated virtual setups to simulate the tooth movements of the presurgical orthodontic treatment. Prior to surgery, digital dental models were merged with the CBCT scans. Differences between de virtual setups and the presurgical dental models were calculated using linear mixed model analyses. Differences in tooth displacements exceeding the boundaries of clinical acceptance (>2 degrees for rotations and >0.6 mm for translations) were found in 75% of the rotational and 52% of translational mean differences in the maxilla and in 74% of the rotational mean differences and 44% of the translational mean differences in the mandible. Significant differences were found for all tooth types and in all tooth displacement directions with significant effects of extractions and surgically assisted rapid maxillary expansion (SARME) procedures. The accuracy of the digital setup is still too limited to correctly simulate the presurgical orthodontic treatment.

## 1. Introduction

Diagnostic orthodontic setups have been used for decades to simulate different treatment options during orthodontic planning. Diagnostic setups aid decision making during treatment planning, for example considering the use of interproximal reduction (IPR), extraction therapy or anchorage management [1,2]. They can also be used to enhance patient communication. The introduction of different 3D imaging and scanning techniques allowed a fully digitized orthodontic treatment planning workflow [3]. Digitalization of the setups offers new opportunities compared to the original manual setups like superimposition of the setup with the original models, easy transferability and duplication. Digital setups are nowadays also frequently used for therapeutic purposes during planning of aligner therapy and other digital techniques as customized wires and buccal and lingual brackets [4,5,6].

Recently a new application of a diagnostic setup in orthognathic treatment planning was introduced [7]. At the start of treatment a diagnostic setup is created. This setup is merged with the corresponding cone beam computed tomography (CBCT) scan. After the pre-surgical orthodontic treatment phase, CBCT imaging and digital dental impressions of the pre-surgical treatment outcome are obtained. Both CBCT scans are superimposed using voxel-based matching on an unaltered subvolume that consists of the cranial base, forehead, and zygomatic arches. Using an iterative closest point algorithm individual tooth position differences between pre-treatment setup and pre-surgical orthodontic treatment outcome can then be evaluated in three dimensions. With the use of this method, it is possible to evaluate the accuracy of the pre-treatment setup in three-dimensions, allowing for a more predictable orthognathic planning at the start of treatment. Proper orthodontic planning and preparation are essential to the outcome of the whole orthognathic treatment. Incorrect decompensation of the dental arches could lead to inability to correct fully for the skeletal discrepancy [8]. The use of digital pretreatment setups could offer unique benefits in terms of predictability of the orthognathic planning ahead of the whole treatment, which may enhance the treatment efficiency and hence diminish the negative influence on oral health related quality of life of the patient that has been reported for the presurgical orthodontic phase [9]. It also could support patients and orthodontists in the treatment plan decisions.

A study of Hou et al. investigated whether viewing of digital setups influenced orthodontic treatment planning decision or orthodontists’ confidence in their selected outcome [4]. After viewing the setups, a significant change in the treatment plan occurred in 24% of the cases, for example a change of extraction patterns. The use of setups was also associated with higher levels of confidence in the selected plans [4].

Several studies are available concerning the accuracy of therapeutic setups. Two studies looked into the predictability of a setup regarding lingual appliances [10,11], another two studies investigated the accuracy of aligner therapy in achieving the pre-treatment setup tooth position [12,13]. These studies provided interesting insights in the predictability of the used orthodontic appliances, however, only data in two-dimensions were used and no studies concerned the use of regular labial fixed appliances.

Although digital setups in orthognathic treatment planning are regarded as useful tools and a potential advancement to establish a definitive treatment plan, questions remain regarding the accuracy and predictability of the pre-treatment diagnostic setup compared to the actual three-dimensional outcome of the presurgical orthodontic treatment just prior to surgery. The predictability of the presurgical orthodontic treatment stage has not attracted much attention so far. Therefore, the goal of this study is to verify whether pre-treatment digital setups can accurately predict the pre-surgical tooth positions in a 3-dimensional way.

## 2. Materials and Methods

### 2.1. Participants

Twenty-six consecutive patients who underwent a combined orthodontic-orthognathic treatment at the Radboud University Medical Center, department of Dentistry, section of Orthodontics and Craniofacial Biology were selected for this retrospective study. The study was conducted between September 2020 and July 2021. Inclusion criteria were skeletally maturity based on cervical vertebral maturation [14], presence of a skeletal discrepancy requiring combined orthodontic-orthognathic surgical treatment, and the availability of two CBCT scans and two 3D digital models of the dental arches, one prior to orthodontic treatment and one prior to orthognathic surgery. Patients with orofacial clefts and craniofacial anomalies or patients missing more than one tooth per quadrant were excluded.

All patients were treated orthodontically with labial fixed appliances including metal bidimensional self-ligating brackets with Roth prescription (Experience, GC Orthodontics, Breckerfeld, Germany).

The sample size was calculated using the formula described by Pandis [15], considering a power of 90% and an α of 0.05 to detect a difference of 0.6 mm between setup and outcome with a standard deviation of 0.6 mm. The sample size calculation, based on a pilot study with ten patients, showed that at least 21 patients were needed. All patient data was anonymized and de-identified prior to analysis, and all participants had given informed consent. The Institutional Review Board of the Radboud University Medical Centre issued an approval for this investigation (2016–2690).

### 2.2. Image Acquisition

Two CBCT scans were acquired for each patient as part of the treatment protocol for orthognathic surgery cases. One CBCT scan was acquired before the start of the orthodontic treatment and the second CBCT scan was taken four weeks prior to orthognathic surgery. On both occasions, an extended-height CBCT scan was acquired (FOV: 16 × 22 cm, scanning time 2 × 20 s, voxel size 0.4 mm, 3D Imaging System, Imaging Sciences International Inc., Hatfield, PA, USA). Directly after the acquisition of the CBCT scans plaster models of the dental arches were acquired after alginate impression taking. The plaster models were digitized with a laser scanner (R500 3D Dental Laser Scanner, 118 3Shape^®^, Copenhagen, Denmark).

### 2.3. Creating the Orthodontic Virtual Setup

OrthoAnalyzer (3Shape^®^, Copenhagen, Denmark) software was used to produce a virtual orthodontic setup on the pre-treatment dental model. The orthodontist (OdW), who created the virtual setups, had access to all patient’s pretreatment records and the treatment plan, but was blinded to the outcome of the actual orthodontic treatment. Creating a virtual setup started with the determination of arch form and tooth axis. Each of the individual teeth were semi-automatically segmented using the OrthoAnalyzer software [7]. After this stage the teeth were manually repositioned to their ideal position according to the treatment plan and key principles of occlusion: correct molar relationship (in final jaw position), correct crown angulation, correct crown inclination, no rotations, no interdental spaces, an appropriate plane of occlusion, correct interproximal contact points, a normal overjet and overbite (1–4 mm) and midlines and arch shapes according to the original treatment plan (Figure 1C). The original mandibular inter-canine distance was maintained and acted as a guide for obtaining the final maxillary arch width and shape.

### 2.4. Treatment Evaluation

Treatment evaluation was carried out by the method of Baan et al. [7], this method is validated and shows small method errors with intraclass correlation coefficients higher than 0.81 and maximum interobserver mean differences of 0.36 mm for translational differences and 1.54° for rotational differences. The method uses three steps to evaluate the difference in tooth position on the virtual setup compared to the position in the outcome for each tooth independently. After superimposition of the dental models to their corresponding CBCT with IPS CaseDesigner, version 2.2.4 (KLS Martin Group, Tuttlingen, Germany) the 3D rendered skulls with dentitions were imported into the 3DMedX^®^ software, version 1.2.17.0 (3D Lab Radboudumc, Nijmegen, The Netherlands). IPS CaseDesigner automatically aligns the STL files of the dental models with the CBCT scans after indicating the right and left condyle, mesial cusp of the first upper right and left molar and the middle of the two upper incisors. This step is validated and shows a high level of accuracy and agreement [16] (step 1). Voxel-based matching (VBM) in IPS CaseDesigner was used to register the pre-surgery CBCT to the pre-treatment CBCT. The stable regions of the anterior cranial base were used for VBM, registration was done using a validated method [17] (step 2). Figure 1 shows all the necessary digital models in this procedure.

In the 3DMedX^®^ software each individual tooth was rotated and translated from the pre-treatment model to the virtual setup and to the final outcome (step 3). All translations and rotations were recorded and saved for each individual tooth (Figure 2). The six degrees of freedom (DOF) were computed: motion around vertical axis (yaw), motion around sagittal axis (roll), motion around transversal axis (pitch), lateral to medial translations (X), anterior to posterior translations (Y) and cranial to caudal translations (Z).

### 2.5. Statistical Analysis

All statistical analyses were performed using R, version 4.1.3(R Core Team, Vienna, Austria). Linear mixed model analyses were used to calculate the differences between the virtual setup and the actual outcome and the estimation of the fixed effects of SARME and extraction on the result per tooth taking individual patients into account. Linear mixed models were used because of data on tooth level being clustered within patients.

## 3. Results

Twenty-six patients were enrolled in this study, 17 females (65%) and 9 males (35%). The mean age at start was 27 (range 18–51) years. Seventeen patients had Class II/1 malocclusion, three patients had a Class II/2 malocclusion and six patients had Class III malocclusion. Fifteen patients were treated with surgically assisted rapid maxillary expansion (SARME) using a tooth-born Hyrax expansion appliance at the start of treatment to expand the maxillary arch. Twenty-five patients were prepared for bimaxillary osteotomy and one patient for a le-fort-1 osteotomy. In thirteen patients teeth were extracted in order to gain space for the correction of crowded or proclined teeth. Twelve of them had two premolar extractions in the lower arch and one patient had first molar extractions in the mandible and only four of those thirteen patients also had two premolars in the upper arch extracted. One patient had a SARME procedure and extraction therapy in the upper arch before orthodontic treatment. The mean pre-surgical orthodontic treatment time was 19 (range 7–32) months.

### 3.1. Setup Accuracy

The evaluate the accuracy of the orthodontic setup, the virtual setup and the actual outcome of the orthodontic surgical preparation of each patient were compared, based on the registration with the underlying CBCT scans. The differences between setup and outcome for each tooth type per parameter are listed in Table 1 and Figure 3 and Figure 4.

Roll is described as rotation around the sagittal axis. A positive value means lingual root rotation (torque) for lateral teeth and mesial root angulation for frontal teeth. Pitch is described as rotation around the transversal axis. A positive value means backward root rotation. Rotation around the vertical axis is described as Yaw. A positive value means mesial out rotation. X is described as lateral to medial displacement. A positive value means lateral movement, a negative value means medial movement. Y is described as anterior to posterior displacement. A positive value means posterior movement, a negative value means anterior movement. Cranial to caudal displacement is described as Z. A positive value means cranial movement, a negative value means caudal movement.

Table 1 and Figure 3 show statistically significant differences for 13 of the 24 outcomes in the maxilla. The largest mean differences were found for rotational movements of the canines (range −5.69°–2.51°). Yaw displacements showed the largest mean differences (range −5.69°–1.96°) of all degrees of freedom. Translational movements for molars showed the largest mean differences (range −0.38–0.54 mm). Furthermore, relatively larges differences were found for Z displacements (range −1.01–0.31 mm).

Table 1 and Figure 3 also show significant differences for 9 of the 24 outcomes in the mandible. The largest mean differences for translational movements were found for molars (range 0.15–0.29 mm). The left-right displacements, as expressed by X, shows the largest mean differences (range 0.06–0.27 mm). The largest mean differences for rotational movements were found for canines (range −3.90°–−2.48°). Roll displacements showed the largest mean differences of all tooth displacement types (range −2.77°–4.73°). Figure 4 Shows the percentage out of range differences in the maxilla for translations and rotations if limits are set at 0.6 mm for translations and 2 degrees for rotations [11,18,19] For both mandible and maxilla a clear difference is seen between translational and rotational differences, with a high percentage out of range differences for rotations and a relatively low percentage of out-of-range differences for translations.

### 3.2. Effect of SARME and Extractions on the Results

Table 2 shows the effects of SARME on the differences between setup and outcome. Significant negative values are found for the pitch for all maxillary tooth types and for the mandibular incisors. Significant effects were found for anterior to posterior movements, as expressed by Y, for the canines, premolars and molars.

Table 3 shows the effect of extraction in the mandible. Significant negative values for rotations were found for the pitch of incisors and canines in both jaws and the roll of premolars and molars in both jaws as well. Significant negative values for translations were found for the X values for premolars in the maxilla and premolars and molars in the mandible. Significant negative values were found for z values for premolars in the maxilla and mandible. The effect of premolar extractions in the maxilla is not mentioned because of the small number of patients (*n* = 4).

## 4. Discussion

### 4.1. Interpretation of Study Results

In this retrospective study, the accuracy of pre-treatment digital setups of twenty-six patients, planned for orthognathic surgery, was evaluated. The results reveal significant differences between the setups and the actual outcome for all tooth types and in all tooth displacement directions indicating a discrepancy between expected tooth movements and the outcome. Incisors, canines and premolars in the maxilla showed significantly more extrusion than expected in the setup. Upper and lower canines showed more mesial crown angulation, backward and distal crown rotation. Lower canines were more buccally positioned. Premolars and molars in the maxilla had a more posterior position in the actual outcome. Premolars showed more distal rotation in both jaws and more buccal and backward crown rotation in the maxilla. Lower premolars were more buccally positioned. Upper molars did not reach the expected expansion as planned in the setup and showed more mesial rotation. In the mandible more buccal crown rotation and more extrusion than expected in the setup were found for molars.

The differences in this study between setups and outcomes were evaluated in relation to the patient’s facial skeleton due to integration of the dental model in the CBCT scans. This could explain the relatively large differences in vertical dimension in this study.

For an assessment of the clinical relevance of the differences between setup and outcome tolerance levels are considered. Based on available tolerance levels in the literature [11,18,19], limits of 0.6 mm for translations and 2 degrees for rotational differences were chosen. Differences exceeding the boundaries of these limits were found in 75% of the rotational and 52% of translational mean differences in the maxilla (Figure 4A,B). For the mandible 26% of the rotational mean differences and 56% of the translational mean differences fell within the range of clinical acceptance (Figure 4C,D).

There’s a lack of reliable data on this subject, which contrasts with the increasing use of digital setups in daily practice. There are some studies regarding the accuracy of setups in patients planned for clear aligner treatment [12,13] and for lingual orthodontic appliances [10,11]. However, these studies did not consider the pre-treatment orthodontic setup for surgical cases and did not use a CBCT-based 3D approach. This hampers a direct comparison of the results. These previous studies have reported higher accuracies between setup and outcome.

The error of the method in the present study can be regarded as a sum of errors related to the superimposition of the CBCT scans and dental models. The maximum mean error of the method ranged up to 0.36 mm for translational differences and 1.54° for rotational differences.

Another possible explanation could be the use of customized preprogrammed appliances in the aforementioned studies. Although literature is ambiguous, customized lingual appliances could be more accurate in achieving the expected outcome [20]. Concerning orthodontic aligners, low to moderate levels of certainty in efficiency of certain orthodontic aligner tooth movements were identified in a systematic review [21]. Clear aligners may produce clinically acceptable outcomes that could be comparable to fixed appliance therapy for buccolingual inclination of upper and lower incisors in mild to moderate malocclusions, but the precise effectiveness of the use of aligners in complex cases compared to fixed appliances still needs to be investigated [21,22]. Another explanation is the treatment complexity of the included cases compared to patients in the previous studies. In the earlier studies patients with extractions or surgical treatment need were excluded [6,7,8,9]. Complex cases introduce more reliability errors compared to general orthodontic cases during set-up production [18].

Additional analyses were performed to investigate the role of extractions and the SARME procedure on the results. In general, we can conclude that in extraction cases ideal arch shape and torque of incisors were more difficult to maintain or achieve. Furthermore, we see difficulties in maintaining the right angulation of teeth neighboring the extraction diastema and more unwanted deepening of the curve of Spee. These are well known effects of premolar extractions on the orthodontic outcome and were confirmed by the results in this study [23].

In addition, SARME may have played a role. Significant errors were seen for the pitch for all tooth types in the maxilla and incisors in the mandible, as a result more retroclined incisors and more distal tipping of (pre)molars were seen in the setups.

Canines, premolars and molars were significantly more posterior positioned in the outcome than in the setup. Upper molars showed more extrusion than expected. An increased dental show and inferior movement of the maxilla after SARME has been described earlier [24]. This agrees with the effect found in this study and can be interpreted as a true influence on the found differences between setup and outcome.

### 4.2. Limitations of the Study Design

A limitation of this study is that only one operator fabricated the orthodontic setups. We did so, to remove the operator effect. Nevertheless, the method is subject to inter-operator variability. Previously, it was shown that inter-operator differences in producing set-ups were clinically acceptable for 74% to 90% of the differences [18]. Although the operator in the present study was an experienced orthodontist and well known to the used software, results could be influenced by the subjective judgment of the clinician [25]. Furthermore, the inclusion of a mixture of dentofacial deformities, may influence the prediction accuracy of the setups which could vary between deformities. This may affect generalizability of the achieved results. Another limitation is the retrospective design of the study, only patients who continued treatment successfully were included which potentially leads to selection bias. Although all patients were treated in the same clinic, different orthodontists were responsible for individual patients which could lead to clinician bias. The subgroup analyses of the effects of SARME and extractions were made on relatively small sample sizes, and the related interpretations should therefore be taken with caution. Although the effects of extractions and SARME are described in this study, other variables such as the type of skeletal discrepancy, age and sex were not investigated. More research is needed to explore the role of those variables.

### 4.3. Clinical Relevance

Although the results in this study indicate that the use of a digital orthodontic setup to predict the pre-surgical outcome of orthodontic treatment is not yet accurate enough, especially in extraction and SARME cases, more research can improve the accuracy. Method errors can be diminished by using more accurate algorithms and artificial intelligence and automatization of the procedure can reduce the operator susceptibility. Use of customized preprogrammed appliances based on the setups instead of the use of non-customized appliances might also lead to a more predictable outcome of the presurgical orthodontic treatment. 

## 5. Conclusions

For certain tooth movements, there are significant differences between the orthodontic setup and the actual outcome of the presurgical orthodontic treatment as part of an orthodontic-orthognathic treatment trajectory. The accuracy of the digital setup is still too limited to correctly simulate the outcome of presurgical orthodontic treatment.

## 6. Patents

No patents resulted from the work reported in this manuscript.

## Figures and Tables

**Figure 1 jcm-11-06141-f001:**
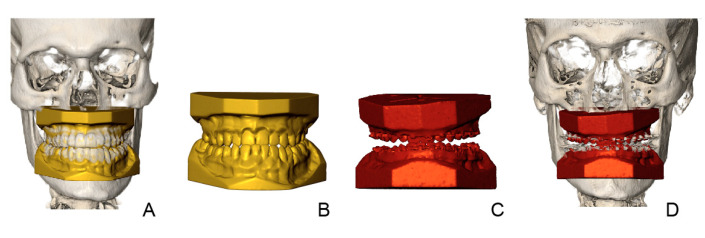
Patient with Angle Class III malocclusion, planned for bimaxillary osteotomy. Overview of all data. (**A**) Virtual setup in pre-treatment CBCT. (**B**) Virtual setup (**C**) pre-surgery model (**D**) pre-surgery model in corresponding CBCT.

**Figure 2 jcm-11-06141-f002:**
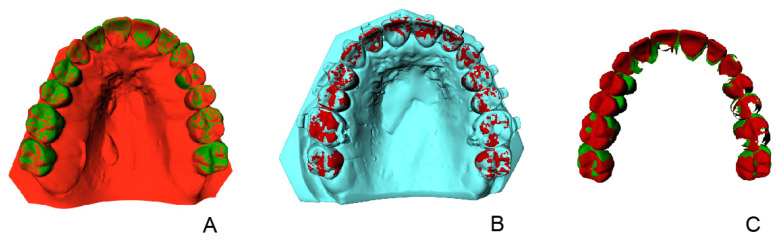
Overview of movement of segmented teeth from (**A**) virtual setup to (**B**) pre-surgery model. (**C**) superposition of segmented teeth of the virtual setup and actual presurgical model. All movements are digitally recorded to enable calculation of the differences between virtual setup and outcome.

**Figure 3 jcm-11-06141-f003:**
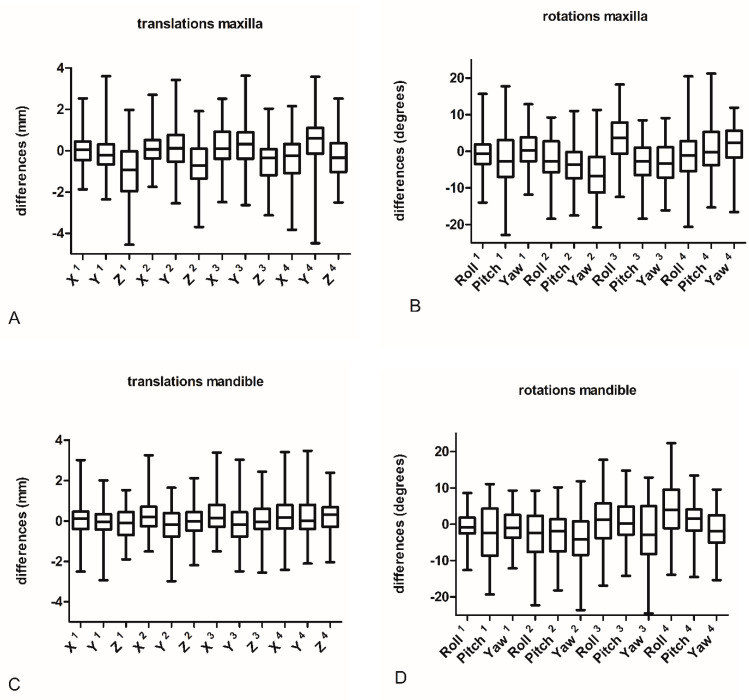
Boxplot display of differences between virtual setup and outcome for maxilla (**A**,**B**) and mandible (**C**,**D**) for translations and rotations. ^1^ incisors, ^2^ canines, ^3^ premolars, ^4^ molars.

**Figure 4 jcm-11-06141-f004:**
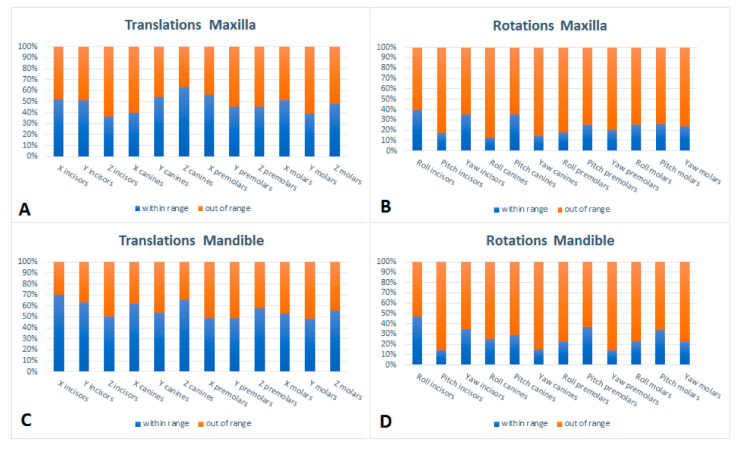
Out of range differences (percentages) for the maxilla (**A**,**B**) and mandible (**C**,**D**). Limits are set at 0.6 mm for translations and 2 degrees for rotations.

**Table 1 jcm-11-06141-t001:** Mean differences between virtual setup and outcome for maxilla and mandible, 95% confidence interval (CI) and results of the linear mixed model analyses (*p* value).

Tooth Type	Parameter	*N*	Mean	95% CI		*p*
				Upper	Lower	
MAXILLA						
*incisors*	Roll	100	−0.64	−1.85	0.58	0.303
	Pitch	100	−1.98	−5.06	1.11	0.208
	Yaw	100	0.34	−0.67	1.38	0.500
	X	100	0.04	−0.10	0.19	0.567
	Y	100	−0.02	−0.46	0.41	0.914
	Z	100	−1.01	−1.58	−0.44	**0.001**
*canines*	Roll	52	−2.51	−4.65	−0.37	**0.021**
	Pitch	52	−4.18	−6.10	−2.26	**0.000**
	Yaw	52	−5.69	−8.16	−3.22	**0.000**
	X	52	0.08	−0.18	0.33	0.552
	Y	52	0.25	−0.11	0.61	0.173
	Z	52	−0.70	−1.16	−0.25	**0.002**
*premolars*	Roll	84	3.42	1.51	5.30	**0.000**
	Pitch	84	−2.96	−4.54	−1.40	**0.000**
	Yaw	84	−3.08	−4.73	−1.46	**0.000**
	X	84	0.20	−0.15	0.54	0.271
	Y	84	0.38	0.01	0.74	**0.043**
	Z	84	−0.47	−0.83	−0.10	**0.011**
*molars*	Roll	99	−0.76	−3.16	1.65	0.537
	Pitch	99	0.88	−1.17	2.91	0.403
	Yaw	99	1.96	0.48	3.45	**0.009**
	X	99	−0.38	−0.76	−0.01	**0.048**
	Y	99	0.54	0.21	0.86	**0.001**
	Z	99	−0.31	−0.64	0.02	0.062
*MANDIBLE*						
*incisors*	Roll	104	−0.66	−1.64	0.33	0.190
	Pitch	104	−2.82	−5.59	−0.05	**0.046**
	Yaw	104	−0.71	−1.67	0.25	0.148
	X	104	0.06	−0.10	0.23	0.432
	Y	104	−0.04	−0.35	0.26	0.778
	Z	104	−0.11	−0.41	0.18	0.454
*canines*	Roll	52	−2.77	−4.92	−0.62	**0.011**
	Pitch	52	−2.48	−4.50	−0.45	**0.016**
	Yaw	52	−3.94	−5.92	−1.96	**0.000**
	X	52	0.26	0.02	0.50	**0.035**
	Y	52	−0.16	−0.48	0.16	0.326
	Z	52	0.03	−0.25	0.31	0.819
*premolars*	Roll	73	1.34	−1.26	3.87	0.318
	Pitch	73	0.72	−1.05	2.48	0.426
	Yaw	73	−2.29	−4.15	−0.47	**0.014**
	X	73	0.27	0.03	0.49	**0.024**
	Y	73	−0.08	−0.43	0.27	0.661
	Z	73	0.04	−0.28	0.36	0.819
*molars*	Roll	93	4.73	1.53	7.90	**0.004**
	Pitch	93	1.15	−0.07	2.38	0.063
	Yaw	93	−1.83	−3.48	−0.20	0.028
	X	93	0.21	−0.07	0.49	0.146
	Y	93	0.15	−0.18	0.49	0.366
	Z	93	0.29	0.02	0.56	0.036

Statistically significant values are marked in bold. Rotational movements are in degrees and translational movements are in mm. Significance was set a α = 0.05.

**Table 2 jcm-11-06141-t002:** Effect of SARME (*N* = 15) on the differences between virtual setup and outcome per jaw, estimate of the effect, 95% confidence interval (CI) and results of the linear mixed model analyses (*p* value) per jaw.

Tooth Type	Parameter	Estimate	95% CI		*p*	Estimate	95% CI		*p*
			Upper	Lower			Upper	Lower	
		MAXILLA				MANDIBLE			
*incisors*	Roll	1.13	−1.66	3.92	0.428	−1.15	−3.92	1.63	0.418
	Pitch	−8.58	−11.10	−6.06	**<0.001**	−7.27	−9.78	−4.77	**<0.001**
	Yaw	−0.70	−2.69	1.29	0.491	−0.51	−2.49	1.46	0.611
	X	0.18	−0.25	0.61	0.403	0.11	−0.32	0.54	0.610
	Y	0.41	−0.08	0.91	0.103	0.17	−0.33	0.66	0.509
	Z	0.22	−0.28	0.71	0.386	−0.33	−0.83	0.16	0.186
*canines*	Roll	−0.24	−3.58	3.09	0.886	−2.52	−5.85	0.82	0.139
	Pitch	−3.87	−6.85	−0.89	**0.011**	−2.57	−5.54	0.41	0.091
	Yaw	1.47	−1.07	4.00	0.256	1.65	−0.88	4.19	0.201
	X	0.26	−0.23	0.75	0.303	0.18	−0.40	0.67	0.459
	Y	0.57	0.02	1.12	**0.042**	0.32	−0.23	0.87	0.248
	Z	0.09	−0.46	0.64	0.747	−0.46	−1.01	0.09	0.102
*premolars*	Roll	0.07	−2.88	3.025	0.961	−2.20	−5.19	0.79	0.149
	Pitch	−3.56	−6.21	−0.90	**0.008**	−2.25	−4.94	0.43	0.100
	Yaw	0.79	−1.36	2.94	0.471	0.98	−1.21	2.49	0.382
	X	0.04	−0.40	0.49	0.845	−0.03	−0.48	0.42	0.905
	Y	0.61	0.10	1.12	**0.019**	0.36	−0.15	0.88	0.166
	Z	−0.11	−0.62	0.40	0.678	−0.66	−1.18	−0.14	0.112
*molars*	Roll	2.46	−0.35	5.26	0.086	0.18	−2.65	3.01	0.900
	Pitch	−2.93	−5.46	−0.40	**0.023**	−1.62	−4.17	0.93	0.213
	Yaw	0.49	−1.51	2.49	0.632	0.68	−1.35	2.70	0.514
	X	−0.10	−0.53	0.33	0.661	−0.17	−0.60	0.27	0.446
	Y	0.53	0.03	1.03	**0.036**	0.29	−0.21	0.79	0.261
	Z	0.54	0.05	1.04	**0.032**	−0.01	−0.51	0.49	0.974

Statistical significant values are marked in bold. Rotational movements are in degrees and translational movements are in mm. Significance was set a α = 0.05.

**Table 3 jcm-11-06141-t003:** Effect of extractions in the mandible (*N* = 13) on the differences between virtual setup and outcome per jaw, estimate of the effect, 95% confidence interval (CI) and results of the linear mixed model analyses (*p* value).

Tooth Type	Parameter	Estimate	95% CI		*p*	Estimate	95% CI		*p*
			Upper	Lower			Upper	Lower	
		MAXILLA				MANDIBLE			
*incisors*	Roll	1.15	−1.62	3.91	0.417	0.38	−2.36	3.12	0.785
	Pitch	−5.80	−8.76	−2.83	**<0.001**	−5.13	−8.09	−2.19	**<0.001**
	Yaw	−0.29	−2.44	1.87	0.794	−1.10	−3.23	1.04	0.313
	X	−0.07	−0.51	0.37	0.754	0.16	−0.29	0.60	0.489
	Y	−0.23	−0.81	0.34	0.421	0.01	−0.56	0.58	0.981
	Z	−0.41	−0.91	0.09	0.110	−0.42	−0.92	0.08	0.102
*canines*	Roll	−1.16	−4.55	2.23	0.503	−1.92	−5.32	1.47	0.266
	Pitch	−4.52	−7.95	−1.08	**0.010**	−3.86	−7.29	−0.42	**0.028**
	Yaw	0.59	−2.14	3.34	0.670	−0.21	−2.96	2.52	−0.877
	X	−0.21	−0.72	0.29	0.408	0.01	−0.50	0.52	0.963
	Y	−0.13	−0.75	0.50	0.691	0.12	−0.51	0.74	0.717
	Z	−0.47	−1.05	0.10	0.102	−0.48	−1.05	0.09	0.099
*premolars*	Roll	−2.95	−5.85	−0.05	**0.046**	−3.71	−6.74	−0.69	**0.016**
	Pitch	−2.37	−5.44	0.69	0.128	−1.71	−4.87	1.44	0.287
	Yaw	1.23	−1.04	3.51	0.288	0.42	−1.97	2.82	0.729
	X	−0.52	−0.98	−0.06	**0.026**	−0.29	−0.76	0.18	0.223
	Y	−0.05	−0.63	0.53	0.870	0.19	−0.40	0.79	0.522
	Z	−0.69	−1.21	−0.18	**0.009**	−0.70	−1.12	−0.17	**0.010**
*molars*	Roll	−3.17	−5.97	−0.38	**0.026**	−3.94	−6.76	−1.12	**0.006**
	Pitch	−2.65	−5.64	0.33	0.082	−1.99	−4.99	1.01	0.194
	Yaw	−0.97	−3.16	1.21	0.381	−1.78	−4.00	−0.42	0.113
	X	−0.68	−1.13	−0.23	**0.003**	−0.46	−0.91	−0.01	**0.047**
	Y	0.27	−0.30	0.85	0.350	0.51	−0.06	1.09	0.079
	Z	−0.39	−0.90	0.12	0.132	−0.40	−0.91	0.11	0.127

Statistical significant values are marked in bold. Rotational movements are in degrees and translational movements are in mm. Significance was set a α = 0.05.

## Data Availability

The data presented in this study are available on request from the corresponding author. The data are not publicly available due to privacy restrictions.

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
