# Peer review of "The Prediction Accuracy of Digital Orthodontic Setups for the Orthodontic Phase before Orthognathic Surgery"

_jcm, 2022, doi:10.3390/jcm11206141_

Round 1
Reviewer 1 Report
The studies was conducted based on accurate superimposition. It's a good topic. Please see my comment below.
I suggest that authors should address the way to superimpose dental model with CBCT in detail. For example, did they superimpose based on landmarks? If so, which landmarks they used? How did they evaluate the results of superimposition? Any reliability of the superimposition result?
Author Response
The studies was conducted based on accurate superimposition. It's a good topic. Please see my comment below.
1. I suggest that authors should address the way to superimpose dental model with CBCT in detail. For example, did they superimpose based on landmarks? If so, which landmarks they used? How did they evaluate the results of superimposition? Any reliability of the superimposition result?
Response 1:
IPS CaseDesigner automatically aligns the STL files of the dental models with the CBCT scans after indicating the right and left condyle, mesial cusp of the first upper right and left molar and the middle of the two upper incisors. This step is validated and shows a high level of accuracy and agreement [15]. We added this to the text of the revised manuscript in paragraph 2.4.

Reviewer 2 Report
The purpose of this study was to verify whether pre-treatment digital setups can accurately predict the tooth positions after presurgical orthodontic treatment have been performed in 3D. The authors used linear mixed model analyses to evaluate the differences between virtual setups and the presurgical outcomes. They found that significant differences between predicted teeth positions in pre-treatment setups and presurgical orthodontic treatment in both mandible and maxilla. The topic greatly attracts readers attention.
Main concern from the reviewer is that how reliable and repeatable of VBM to superimpose the pre-surgery CBCT to the pre-treatment CBCT. Please indicate the source of the VBM, eg. IPS CaseDesigner, version 2.2.4 (KLS Martin Group, 144 Tuttlingen, Germany). Minor changes of the superimposing outcome can deeply affect the quantified results more than 0.5mm in translation or 2 degrees in rotation which are the margin of clinical acceptance.
In abstract, LINEs 28-32, the descriptions in percentages of rotational and translational differences within or exceeding clinical acceptance should be consistency in both maxilla and mandible.
A power of 90% for sample size calculation shows that at least 21 patients were needed. However, 26 patients enrolled in this study in which 15 patients were treated SARME, 13 patients’ teeth were extracted for gaining space of arch. Sample size of both subgroups didn’t meet the requirement of a 90% power. Will it be possibly bias conclusions caused by lack of enough samples?
Fifteen patients, over half of the total patients, underwent SARME procedure. Please describe whether the drawing margin lines and the medical device used in SARME affect the shape of zygomatic bone? If so, the outcomes of teeth differences between pre-treatment setups and presurgical orthodontic treatment were unreliable because the superimposing region in cranial base will be bias.
The descriptions of translation and rotation of tooth are confusion, and need to be corrected. In medical imaging device, we usually applied right-handed coordinate system. The coordinate system in computer tomography, for example, the x, y, z axes define the directions from right to left, anterior to posterior, and caudal to cranial of the subject respectively. Follow the right-handed rule, the pitch, yaw, roll are the positive and negative angles of rotation along x, y, z axes, respectively. Please correct.
There are many duplicate notes below four tables. These notes should be corrected and reorganized described in context before Table 1. Note descriptions below four tables should be removed.
Column title in Tables 3A and 3B should be reformatted.
There are several “star” symbols in Fig.3 and 4 to indicate incisors, canines, premolars, and molars, which were confused with the symbols represented the degree of statistic significant. Please change “star” symbols to numbers.
Subtitle of each subfigure in Fig. 3 and 4 should be changed to a more appropriate word.
Author Response
The purpose of this study was to verify whether pre-treatment digital setups can accurately predict the tooth positions after presurgical orthodontic treatment have been performed in 3D. The authors used linear mixed model analyses to evaluate the differences between virtual setups and the presurgical outcomes. They found that significant differences between predicted teeth positions in pre-treatment setups and presurgical orthodontic treatment in both mandible and maxilla. The topic greatly attracts readers attention.
2. Main concern from the reviewer is that how reliable and repeatable of VBM to superimpose the pre-surgery CBCT to the pre-treatment CBCT. Please indicate the source of the VBM, eg. IPS CaseDesigner, version 2.2.4 (KLS Martin Group, 144 Tuttlingen, Germany). Minor changes of the superimposing outcome can deeply affect the quantified results more than 0.5mm in translation or 2 degrees in rotation which are the margin of clinical acceptance.
Response 2:
Voxel based image registration on anterior cranial base can be considered as an accurate and a reproducible method for CBCT superimposition. We used a validated method as concluded in the article of Nada et al. [Nada et al 2011). For anterior cranial base registration a mean distance between the CBCT’s of 0.2 to 0.37 mm (SD 0.08-0.16) could be expected.
We add the reference to the text of the revised manuscript in paragraph 2.4.
3. In abstract, LINEs 28-32, the descriptions in percentages of rotational and translational differences within or exceeding clinical acceptance should be consistency in both maxilla and mandible.
Response 3:
We agree with your comment. We have added the translational and rotational differences for the mandible, see text changes in the abstract
4. A power of 90% for sample size calculation shows that at least 21 patients were needed. However, 26 patients enrolled in this study in which 15 patients were treated SARME, 13 patients’ teeth were extracted for gaining space of arch. Sample size of both subgroups didn’t meet the requirement of a 90% power. Will it be possibly bias conclusions caused by lack of enough samples?
Response 4:
We agree that conclusions should be taken with caution, we added a sentence in paragraph 4.2 (limitations of the study design) concerning this limitation: ‘’The subgroup analyses of the effects of SARME and extractions were made on relatively small sample sizes, and the related interpretations should therefore be taken with caution’’.
5. Fifteen patients, over half of the total patients, underwent SARME procedure. Please describe whether the drawing margin lines and the medical device used in SARME affect the shape of zygomatic bone? If so, the outcomes of teeth differences between pre-treatment setups and presurgical orthodontic treatment were unreliable because the superimposing region in cranial base will be bias.
Response 5:
According to literature [Nada 2012] this registration step is reliable and there are only differences seen in the anterior maxillary area after expansion using our tooth-born expansion appliance. We didn’t superimpose on these areas nor on the anterior maxillary region nor on the zygomatic bone, we corrected this in text of the revised manuscript in paragraph 2.4. We used a tooth-borne hyrax appliance. We added this in paragraph 3.
6. The descriptions of translation and rotation of tooth are confusion, and need to be corrected. In medical imaging device, we usually applied right-handed coordinate system. The coordinate system in computer tomography, for example, the x, y, z axes define the directions from right to left, anterior to posterior, and caudal to cranial of the subject respectively. Follow the right-handed rule, the pitch, yaw, roll are the positive and negative angles of rotation along x, y, z axes, respectively. Please correct.
Response 6:
We agree that in other context your description is correct.
We changed this because this is contrast to the ‘orthodontic/dental’ language in literature. For example left movement of molar 16 is an opposite lingual move compared to left/buccal movement of molar 26. Therefore we mirrored data and changed the description to describe the tooth movements appropriately and understandable to dental professionals.
7. There are many duplicate notes below four tables. These notes should be corrected and reorganized described in context before Table 1. Note descriptions below four tables should be removed.
Response 7:
We removed the duplicate notes and described them in context in paragraph 3.1 before table 1.
8. Column title in Tables 3A and 3B should be reformatted.
Response 8:
we corrected column titles.
9. There are several “star” symbols in Fig.3 and 4 to indicate incisors, canines, premolars, and molars, which were confused with the symbols represented the degree of statistic significant. Please change “star” symbols to numbers.
Response 9:
We changed ‘star symbols” into numbers for fig. 3, another reviewer preferred full written so we changed “star symbols” for fig..4 into tooth types.
10. Subtitle of each subfigure in Fig. 3 and 4 should be changed to a more appropriate word.
Response 10:
We think these titles are clear and appropriate.

Reviewer 3 Report
Thanks for inviting me to review this work.
The study evaluated the 3D prediction accuracy of the presurgical orthodontic preparation phase. I have the following issues that need to be addressed.
Title
1- The title, in its current form, does reflect the content of the paper. The suggested title should be: "The prediction accuracy of digital orthodontic setup for the preparatory orthodontic phase before orthognathic surgery." The authors should clarify that they studied the prediction accuracy of the first phase of orthognathic procedures. The current title is generic and not specific.
Abstract
2- The word 'outcomes' may indicate the studied variables or the final presurgical model of the actual orthodontic tooth movements. Here, I hope that authors will avoid using the word outcome to indicate the actual final presurgical dental model. Outcomes are generally used to express the variables measured or calculated. Please correct this in the Abstract section.
Introduction
3- In the first paragraph, Line 47, a sentence should be added about the introduction of different 3D imaging and scanning techniques in recent years that allowed the workflow in orthodontic treatment planning to become fully digitized (i.e., before mentioning the benefits of using digitized setups in the following sentence). Also, I prefer to support this added sentence with the following citation:
Hajeer MY, Millett DT, Ayoub AF, Siebert JP. Applications of 3D imaging in orthodontics: part II. J Orthod. 2004 Jun;31(2):154-62. doi:10.1179/146531204225020472. PMID: 15210932.
4- In the last sentence of the first paragraph, the authors say that digital setups are nowadays frequently used for planning procedures in aligner therapy and other techniques such as customized wires and brackets. Please add "buccal and lingual" before "brackets" to indicate the possible uses in vestibular and lingual techniques. Add this very recent citation to this sentence, please:
Kara-Boulad JM, Burhan AS, Hajeer MY, Khattab TZ, Nawaya FR, Al-Sabbagh R. Treatment of Moderately Crowded Teeth Using Lingual Fixed Appliance Prepared by a Modified HIRO® Technique: A Case Report and Method Description. Cureus. 2022 May 17;14(5):e25077. doi: 10.7759/cureus.25077. PMID: 35600066; PMCID: PMC9117840.
5- The word 'fused' is not bad, but I prefer to use the word 'merge' instead. The method is dependent on merging data coming from the digital setup with the CBCT data.
6- The justification of the onset of this work is not very strong. I hope the authors explain the importance of accurately predicting the presurgical tooth movements and possible untoward effects if the orthodontist fails to accomplish what he/she intended to do before initiating treatment. You must convince the readers that adhering to what was planned orthodontically may greatly affect the overall orthognathic surgery treatment outcome.
Materials and Methods
7- Please expand the exclusion criteria for orthognathic surgery patients. Any Class III and II patient can be considered a "developmental deformity". Please use an appropriate word to describe those patients excluded because of syndromes or congenerical deformities (i.e., cleft lip and palate patients).
8- Here, the use of "bi-dimensional self-ligating brackets" means that the authors employed a system not commonly used in the orthodontic practice, i.e., in other words, this would affect the generalizability of the achieved results. A point that should be commented on in your discussion.
9- How did you obtain consent forms if the study was retrospective?
10- Who did the treatment planning? Is aligning midlines with the facial midline one of the objectives of presurgical orthodontic preparation? Do we need to align the midlines of the dental arches with the facial midline before performing the surgical intervention? Is this the general rule?
11- On page 4, Line 134, what is meant by this sentence "the virtual setup was made also considering treatment factors like the anatomical boundaries, wire play, arch form, and the expected torque loss". I could not understand this sentence. What was made specifically about these factors? Please explain more.
12- Figure 2 caption: please replace the word 'outcome' with the following phrase, "actual presurgical model," since this would make the sentence more understandable.
13- What is meant by 'linear mixed model analysis by restricted maximal likelihood? Please explain more with clearer information.
Results
14- A mixture of dentofacial deformities in this validation work is a problem. Here, we have 26 patients with different types of deformities. We all know that the preparation stages differ from case to case. A mixture of different deformities means that the prediction accuracy is not the same between the different families of malocclusions. In other words, a better design of this study would have included only Class III patients treated by bimaxillary intervention or maybe one-jaw surgery. This should be commented on in your Discussion section as one of the limitations of the current work.
15- The mean presurgical orthodontics treatment was 19 months. This mean value is somewhat high. We usually prepare our patients within 9 to 15 months. Could you please explain this lengthy preparation treatment?
16- Page 5, Line 203: Please change the phrase "all tooth displacement types" to "degrees of freedoms". The latter phrase is wider and includes rotational and displacement movements.
17- "out-of-range" differences had the thresholds of 2 degrees and 0.6 mm. However, in the sample size calculation, you chose a clinically important difference of 0.5 mm. Could you explain why you changed the threshold from 0.5 mm to 0.6 mm?
18- In the legend of Table 1, the word 'motion', which appeared six times, should be replaced by 'rotation' for the roll, pitch and yaw variables, respectively. It should also be replaced by the word 'translation' or 'displacement' for the x, y, and z axes. This also should be done to Tables 2, 3A, and 3B.
19- In Figure 3, please try to delete the asterisks in each graph and put the words 'incisors', 'canines', 'premolars', and 'molars' in their appropriate positions. The same should be done in Figure 4. The use of asterisks in this manner is not favorable.
20- Tables 3A and 3B present the results related to only a few patients (4 patients in Table 3A and 13 patients in Table 3B); therefore, the statistical power is very limited. I strongly suggest deleting those tables. Also, no strong conclusions should be built on underpowered statistical analyses.
Discussion
21- In the limitation section, please state that the subgroup analyses were made on small numbers of patients, i.e. the related interpretations should be taken with caution.
Author Response
Thanks for inviting me to review this work.
The study evaluated the 3D prediction accuracy of the presurgical orthodontic preparation phase. I have the following issues that need to be addressed.
Comments and Suggestions for Authors
Title
1. The title, in its current form, does reflect the content of the paper. The suggested title should be: "The prediction accuracy of digital orthodontic setup for the preparatory orthodontic phase before orthognathic surgery." The authors should clarify that they studied the prediction accuracy of the first phase of orthognathic procedures. The current title is generic and not specific.
Response 1:
Thank you for this remark, which is a very good suggestion. We changed the title accordingly.
Abstract
2. The word 'outcomes' may indicate the studied variables or the final presurgical model of the actual orthodontic tooth movements. Here, I hope that authors will avoid using the word outcome to indicate the actual final presurgical dental model. Outcomes are generally used to express the variables measured or calculated. Please correct this in the Abstract section.
Response 2:
Thank you for your observation. We agree and have corrected this in the revised version of the Abstract section.
Introduction
3. In the first paragraph, Line 47, a sentence should be added about the introduction of different 3D imaging and scanning techniques in recent years that allowed the workflow in orthodontic treatment planning to become fully digitized (i.e., before mentioning the benefits of using digitized setups in the following sentence). Also, I prefer to support this added sentence with the following citation:
Hajeer MY, Millett DT, Ayoub AF, Siebert JP. Applications of 3D imaging in orthodontics: part II. J Orthod. 2004 Jun;31(2):154-62. doi:10.1179/146531204225020472. PMID: 15210932.
Response 3:
As suggested by the refer we added a sentence and the reference in the Introduction section first paragraph.
4. In the last sentence of the first paragraph, the authors say that digital setups are nowadays frequently used for planning procedures in aligner therapy and other techniques such as customized wires and brackets. Please add "buccal and lingual" before "brackets" to indicate the possible uses in vestibular and lingual techniques. Add this very recent citation to this sentence, please:
Kara-Boulad JM, Burhan AS, Hajeer MY, Khattab TZ, Nawaya FR, Al-Sabbagh R. Treatment of Moderately Crowded Teeth Using Lingual Fixed Appliance Prepared by a Modified HIRO® Technique: A Case Report and Method Description. Cureus. 2022 May 17;14(5):e25077. doi: 10.7759/cureus.25077. PMID: 35600066; PMCID: PMC9117840.
Response 4:
We have adjusted the text to enable inclusion of the reference suggested by the referee (Introduction section first paragraph).
5- The word 'fused' is not bad, but I prefer to use the word 'merge' instead. The method is dependent on merging data coming from the digital setup with the CBCT data.
Response 5:
Although not specifically asked, we replaced ‘fused’ by ‘merged’.
6- The justification of the onset of this work is not very strong. I hope the authors explain the importance of accurately predicting the presurgical tooth movements and possible untoward effects if the orthodontist fails to accomplish what he/she intended to do before initiating treatment. You must convince the readers that adhering to what was planned orthodontically may greatly affect the overall orthognathic surgery treatment outcome.
Response 6:
Thank you for your comment this was missing.
We added the following sentences to the Introductions section:
Proper orthodontic planning and preparation are essential to the outcome of the whole orthognathic treatment. Incorrect decompensation of the dental arches could lead to inability to correct fully for the skeletal discrepancy [Larson 2014].
and
The use of digital pretreatment setups could offer unique benefits in terms of predictability of the orthognathic planning ahead of the whole treatment, which may enhance the treatment efficiency and hence diminish the negative influence on oral health related quality of life of the patient that has been reported for the presurgical orthodontic phase (Brouns et al, 2022). It also could support patients and orthodontists in the treatment plan decisions.
Materials and Methods
7- Please expand the exclusion criteria for orthognathic surgery patients. Any Class III and II patient can be considered a "developmental deformity". Please use an appropriate word to describe those patients excluded because of syndromes or congenerical deformities (i.e., cleft lip and palate patients).
Response 7:
We clarified the description of the exclusion criteria to make it more clear (paragraph 2.1): Patients with orofacial clefts and craniofacial anomalies or patients missing more than one tooth per quadrant were excluded.
8- Here, the use of "bi-dimensional self-ligating brackets" means that the authors employed a system not commonly used in the orthodontic practice, i.e., in other words, this would affect the generalizability of the achieved results. A point that should be commented on in your discussion.
Response 8:
We kindly disagree with the reviewer. There is no evidence for differences in quality or outcome between different bracket systems. The treatment plan and the quality demands of the orthodontist are key factors in the outcome. Generalizability is not significantly affected by the use of this less commonly used system. [Fleming 2010]
9- How did you obtain consent forms if the study was retrospective?
Response 9:
All patients sign a standardized informed consent before treatment, including use of anonymous patient material for scientific research. In case material cannot be anonymized, the data used for statistics will be anonymized before further processing. This a standard procedure at the university hospital.
10- Who did the treatment planning? Is aligning midlines with the facial midline one of the objectives of presurgical orthodontic preparation? Do we need to align the midlines of the dental arches with the facial midline before performing the surgical intervention? Is this the general rule?
Response 10:
No it is not a general rule. Planning of the setup was according to treatment plan of the treating orthodontist. To be able to compare the setup with the actual presurgical model, the planning of the midline should be known (i.e., if (no) correction was part of treatment this should be known by the set-up maker). We clarified the text in paragraph 2.3.
11- On page 4, Line 134, what is meant by this sentence "the virtual setup was made also considering treatment factors like the anatomical boundaries, wire play, arch form, and the expected torque loss". I could not understand this sentence. What was made specifically about these factors? Please explain more.
Response 11:
We agree this is confusing. In the revised manuscript (paragraph 2.3) we added arch shape to the former sentence because this is essential for fitting the jaws in the setup. The other factors were removed because they are part of the orthodontic treatment itself and do not need to be mentioned here.
12- Figure 2 caption: please replace the word 'outcome' with the following phrase, "actual presurgical model," since this would make the sentence more understandable.
Response 12:
Thank you for this suggestion. We changed the figure 2 legend accordingly.
13- What is meant by 'linear mixed model analysis by restricted maximal likelihood? Please explain more with clearer information.
Response 13:
We changed the entire paragraph 2.5 (statistical analysis) because it might be difficult to understand for the reader. The text reads now: All statistical analyses were performed using R version 4.1.3). Linear mixed model analyses were used to calculate the differences between the virtual setup and the actual outcome and the estimation of the fixed effects of SARME and extraction on the result per tooth taking individual patients into account. Linear mixed models were used because of data on tooth level being clustered within patients.
Results
14- A mixture of dentofacial deformities in this validation work is a problem. Here, we have 26 patients with different types of deformities. We all know that the preparation stages differ from case to case. A mixture of different deformities means that the prediction accuracy is not the same between the different families of malocclusions. In other words, a better design of this study would have included only Class III patients treated by bimaxillary intervention or maybe one-jaw surgery. This should be commented on in your Discussion section as one of the limitations of the current work.
Response 14:
We added the following sentence to the discussion (paragraph 4.2):
Furthermore, the inclusion of a mixture of dentofacial deformities, may influence the prediction accuracy of the setups which could vary between deformities. This may affect generalizability of the achieved results.
15- The mean presurgical orthodontics treatment was 19 months. This mean value is somewhat high. We usually prepare our patients within 9 to 15 months. Could you please explain this lengthy preparation treatment?
Response 15:
This is possibly due to the academic setting with high quality demands of the result. Furthermore our treatment policy is to make the post-surgical orthodontic phase as short as possible as patient compliance diminishes after surgery. We did not register the post-surgical period in this study.
16- Page 5, Line 203: Please change the phrase "all tooth displacement types" to "degrees of freedoms". The latter phrase is wider and includes rotational and displacement movements.
Response 16:
We corrected for this in text paragraph 3.1
17- "out-of-range" differences had the thresholds of 2 degrees and 0.6 mm. However, in the sample size calculation, you chose a clinically important difference of 0.5 mm. Could you explain why you changed the threshold from 0.5 mm to 0.6 mm?
Response 17:
Thank you very much for detecting this mistake. We did not change the threshold which was 0.6 mm from the beginning. We corrected the values in paragraph 2.1.
18- In the legend of Table 1, the word 'motion', which appeared six times, should be replaced by 'rotation' for the roll, pitch and yaw variables, respectively. It should also be replaced by the word 'translation' or 'displacement' for the x, y, and z axes. This also should be done to Tables 2, 3A, and 3B.
Response 18:
We agree with this and changed this in text in paragraph 3.1. Because of the comments of reviewer 2 we removed this part out of the legends and placed them in context before the tables.
19- In Figure 3, please try to delete the asterisks in each graph and put the words 'incisors', 'canines', 'premolars', and 'molars' in their appropriate positions. The same should be done in Figure 4. The use of asterisks in this manner is not favorable.
Response 19:
For fig. 3 this was unfortunately not possible and therefore in the revised manuscript we changed the asterisks into superscript numbers that are explained below the graphs. For fig. 4. We changed it as suggested.
20- Tables 3A and 3B present the results related to only a few patients (4 patients in Table 3A and 13 patients in Table 3B); therefore, the statistical power is very limited. I strongly suggest deleting those tables. Also, no strong conclusions should be built on underpowered statistical analyses.
Response 20:
We agree table 3A is of limited value and we removed this table.
Table 3B is also of limited value to draw strong conclusions but give a good insight into the diversity of our patient population and we prefer to keep them in the article. We added a sentence to the discussion to emphasize the limited value of this Table ( paragraph 4.2).
Discussion
21- In the limitation section, please state that the subgroup analyses were made on small numbers of patients, i.e. the related interpretations should be taken with caution.
Response 21:
We agree, we made statement in the discussion section (paragraph 4.2) and removed Table 3A.
References:
Nada, R.M., et al., Accuracy and reproducibility of voxel based superimposition of cone beam computed tomography models on the anterior cranial base and the zygomatic arches. PLoSOne, 2011. 6(2): p. e16520.
Nada, R.M., et al., Three-dimensional prospective evaluation of tooth-borne and bone-borne surgically assisted rapid maxillary expansion. J Craniomaxillofac Surg, 2012. 40(8): p. 757-62.
Brouns, V.,et al.,Oral health-related quality of life before, during, and after orthodontic-orthognathic treatment: a systematic review and meta-analysis. Clin Oral Investig 2022 Vol. 26 Issue 3 Pages 2223-2235. Accession Number: 35194682 DOI: 10.1007/s00784-021-04288-7
Larson, B.E., Orthodontic preparation for orthognathic surgery. Oral Maxillofac Surg Clin North Am, 2014. 26(4): p. 441-58.
Fleming, P.S. and A. Johal, Self-ligating brackets in orthodontics. A systematic review. Angle Orthod, 2010. 80(3): p. 575-84.

Round 2
Reviewer 2 Report
No more suggestions.
Reviewer 3 Report
Thanks for addressing all of my comments and suggestions.